# Feasibility of Ex Vivo Ligandomics

**DOI:** 10.3390/biom15010145

**Published:** 2025-01-18

**Authors:** Prabuddha Waduge, Remya Ammassam Veettil, Bojun Zhang, Chengchi Huang, Hong Tian, Wei Li

**Affiliations:** 1Cullen Eye Institute, Department of Ophthalmology, Baylor College of Medicine, Houston, TX 77030, USA; 2LigandomicsRx, LLC, Houston, TX 77098, USA

**Keywords:** ligandomics, ex vivo ligandomics, in vivo ligandomics, Scg3, VEGF, ex vivo ligand binding assay

## Abstract

We developed ligandomics for the in vivo profiling of vascular ligands in mice, discovering secretogranin III (Scg3) as a novel angiogenic factor that selectively binds to retinal vessels of diabetic but not healthy mice. This discovery led to the development of anti-Scg3 therapy for ocular vasculopathies. However, in vivo ligandomics requires intracardial perfusion to remove unbound phage clones, limiting its use to vascular endothelial cells (ECs). To extend ligandomics to non-vascular cells, we investigated ex vivo ligandomics. We isolated ECs and retinal ganglion cells (RGCs) from diabetic and healthy mouse retinas by immunopanning. We quantified the binding of clonal phages displaying Scg3 and vascular endothelial growth factor (VEGF), confirming that their binding patterns to isolated diabetic versus healthy ECs matched in vivo patterns. Additionally, Scg3 and VEGF binding to isolated RGCs reflected their in vivo activity. These results support the feasibility of ex vivo ligandomics. We further mapped ligands binding to immunopanned diabetic and healthy ECs and RGCs by ligandomics, confirming that Scg3 was enriched with selective binding to diabetic ECs but not healthy ECs or diabetic/healthy RGCs. These findings demonstrate the feasibility of ex vivo ligandomics, which can be broadly applied to various cell types, tissues, diseases, and species.

## 1. Introduction

Cell–cell communication is crucial for multicellular organisms to regulate cellular signaling and function, guide development and differentiation, maintain homeostasis, coordinate immune responses and neural interactions, and initiate and mitigate pathological processes [1]. Ligand–receptor interactions (LRIs) on the cell surface are central to this communication, and cellular ligands that connect different cells are critical for delineating disease mechanisms and therapeutic targets [2]. Although the entire human plasma membrane receptome has been fully mapped based on the transmembrane domains and cell surface expression [3], cell-binding ligands are traditionally identified on a case-by-case basis using conventional methods that come with technical challenges. It is even more daunting to discover disease-selective ligands to elucidate disease mechanisms and identify therapeutic targets.

To tackle these challenges, we recently developed ligandomics, a unique technology that globally maps cell-binding ligands and quantifies their binding activity simultaneously [4,5]. We applied ligandomics to mice with or without diabetic retinopathy (DR) or choroidal neovascularization to globally map endothelial ligands binding to ocular vasculatures [4,6]. The quantitative comparison of ligandomics data systematically identified disease-restricted endothelial ligands, including secretogranin III (Scg3) which minimally binds to healthy vasculatures but significantly increases its binding to diseased vessels. In contrast, vascular endothelial growth factor (VEGF) binds equally to both diseased and healthy retinal vessels. Subsequent studies independently validated Scg3 as a disease-selective angiogenic factor and developed Scg3-neutralizing monoclonal antibodies (mAbs) and related humanized antibodies (hAbs) for ligand-guided, disease-targeted anti-angiogenic therapy [7,8]. However, in vivo ligandomics profiling requires the intravenous injection of library clones for vessel binding, followed by intracardial perfusion to remove unbound circulating clones [4,6], limiting its applicability to vascular endothelial cells (ECs) in animal models.

To overcome this limitation, this study investigated the feasibility of ex vivo ligandomics with broad applicability to any cell types and diseases. We isolated ECs and retinal ganglion cells (RGCs) from the retinas of diabetic and healthy mice using immunopanning, quantified the binding of Scg3 and VEGF to isolated cells, and found a high degree of consistency between ex vivo and in vivo binding activity patterns. Additionally, we performed ex vivo ligandomics to systematically identify DR-selective cell-binding ligands. The results suggest that ex vivo ligandomics is applicable to any type of cells, diseases, and species, extending its use beyond in vivo ligandomics in animal models. The potential applications and technical limitations of this new approach are discussed.

## 2. Materials and Methods

### 2.1. Animals and Materials

C57BL/6J male mice, aged 6–8 weeks, were obtained from the Jackson Laboratory. Male mice were selected for this study because of their higher susceptibility to streptozotocin-induced diabetes [9]. The mice were treated with streptozotocin to induce hyperglycemia (blood glucose > 300 mg/dL) and aged for 4 months to develop chronic DR with retinal vascular leakage [4]. All animal procedures were approved by the Institutional Animal Care and Use Committee at Baylor College of Medicine (Protocol # AN-8362; 26 July 2023).

Clonal Scg3-Phage and VEGF-Phage were previously described [6,8,10,11]. The anti-Scg3 humanized antibody (hAb) was generated by Everglades Biopharma (Houston, TX, USA) [12]. Aflibercept, an approved anti-VEGF drug from Regeneron Pharmaceuticals (Tarrytown, NY, USA), was purchased from a pharmacy. Human retinal microvascular endothelial cells (HRMVECs) were purchased from Cell Systems (Kirkland, WA, USA; Cat. #ACBRI 181) and cultured as described [4]. HT22 cells (Sigma, St. Louis, MO, USA; Cat. #SCC129) were cultured in DMEM supplemented with 10% fetal bovine serum, 1× GluMax, and 1× penicillin-streptomycin.

### 2.2. Preparation of Single Retinal Cells

Diabetic and age-matched healthy mice were euthanized by CO_2_ inhalation, followed by intracardial perfusion with 70 mL of phosphate-buffered saline (PBS) for 7 min to remove blood cells. After eye enucleation, retinas were isolated (6 retinas for clonal phage binding experiments and 4 retinas for each round of ex vivo ligandomics enrichment), rinsed with PBS, and incubated with collagenase Type 1-A (1 mg/mL; Sigma, St. Louis, MO, USA; Cat. #C2647) and DNase 1 (0.1 mg/mL; Sigma, Cat. #D4263) in Hanks’ Balanced Salt Solution (HBSS, 0.5 mL). After incubation at 37 °C for 30 min with gentle shaking (~50 rpm) and triturating every 10 min, single cells were filtered through a 40 µm mesh filter (VWR, Radnor, PA, USA; Cat. #89508-342) and centrifuged at 300 g for 5 min. Cell pellets were resuspended in 5 mL or 200 µL of 1× PBS supplemented with 0.5% bovine serum albumin (BSA) and 2 mM EDTA for clonal phage binding or ex vivo ligandomics profiling, respectively.

### 2.3. Immunopanning of Endothelial and Neuronal Cells

Anti-CD31 (Thermo Fisher Scientific, Waltham, MA, USA; Cat. #14-0311-82) and anti-CD90.2 (BioLegend, San Diego, CA, USA; Cat. #105301) mAbs were diluted 1:100 in 50 mM Tris-HCl, pH 9.5, and added to non-tissue-culture-treated 24-well plates (200 µL/well; Corning/Falcon, Durham, NC, USA; Cat. #351147). After incubation at 4 °C overnight, the plates were washed three times with PBS and blocked with PBS containing 0.5% BSA at room temperature (300 µL/well) for 1 h, followed by three additional PBS washes. The single-cell suspension was added to the plates pre-coated with anti-CD31 mAb at 200 µL/well and incubated at 37 °C with gentle shaking every 15 min to capture ECs. After 45 min, the supernatant containing unbound cells was carefully removed, and the plates were washed three times with PBS. The cell suspension removed from the anti-CD31 plates was transferred to the anti-CD90.2 plates, followed by the same procedure to capture RGCs. The captured cells on plates were immediately used for subsequent clonal phage binding quantification or ex vivo ligandomics profiling.

### 2.4. Purity Analysis of Immunopanned Cells

To determine the purity of immunopanned cells, ECs were detached from the plates by pipetting, immunolabeled with DyLight 550-conjugated rat anti-CD31 mAb (1:100 dilution; Novus Biologicals, Centennial, CO, USA; Cat. #NB100-1642R), and incubated for 30 min at 4 °C in the dark. Immunopanned RGCs were detached, fixed with 2% paraformaldehyde for 20 min, washed with PBS containing 0.5% Triton X-100, and immunolabeled with Alexa Fluor 488-conjugated mouse anti-βIII tubulin mAb (1:100 dilution; BioLegend, San Diego, CA, USA; Cat. #657404). After immunolabeling, ECs and RGCs were washed, fixed, rewashed, resuspended in PBS containing 0.5% BSA, and analyzed using a BD LSR II flow cytometer and BD FACS DIVA version 6.0 software (BD Biosciences, Franklin Lakes, NJ, USA). Data were analyzed using FlowJo version 10.

### 2.5. Ex Vivo Ligand Binding Assay

Clonal T7 phage displaying Scg3 (Scg3-Phage) or VEGF (VEGF-Phage) was amplified in BLT5615 bacteria, purified by CsCl gradient centrifugation, dialyzed against PBS, and titrated by a phage plaque assay, as described [6,10,11]. The purified Scg3-Phage or VEGF-Phage (1 × 10^12^ pfu in 200 µL/well) was preincubated with or without anti-Scg3 hAb or aflibercept (4 μg/mL), respectively, in PBS supplemented with 0.5% BSA at room temperature for 30 min, and added to the plates with immunopanned cells. Control phage displaying no foreign protein was included as a negative control. The plates were incubated at 37 °C with 5% CO_2_ for 1 h. Unbound phages were removed by washing with PBS + 0.5% BSA for three times within 5 min incubation per wash. To elute the bound phages, PBS with 1% Triton X-100 was added (75 µL/well) and incubated for 10 min with vigorous shaking to lyse the cells. After mixing with 225 µL/well of LB broth by pipetting, cell-bound phages were quantified by a plaque assay.

### 2.6. Ex Vivo Ligandomics

Comparative ligandomics profiling was carried out as previously described with modifications [4,6,13]. Briefly, open reading frame (ORF) phage display (OPD) cDNA libraries, constructed from mouse adult eyes and E18 embryos, were amplified in BLT5616 *E. coli*, combined to increase gene representation, and purified using CsCl gradient centrifugation, followed by dialysis against PBS. The purified phages (1 × 10^12^ pfu/well) in PBS supplemented with 0.5% BSA were added to anti-CD31-captured ECs or anti-CD90.2-captured RGCs in 24-well plates. After incubation at 37 °C for 1 h, unbound phages were removed by washing with PBS (two, three, and four times for Round 1, 2, and 3, respectively; 1 min each time). To elute the bound phages, PBS with 1% Triton X-100 was added (75 µL/well) and incubated for 10 min to lyse the cells, followed by mixing with 225 µL/well of LB broth. Released phages were quantified by a phage plaque assay, amplified, purified, titrated, and used as input for the next round of ex vivo binding selection. After three rounds of selection, enriched phages were selected for ORF clones by binding to streptavidin pre-immobilized on ELISA plates, washed, eluted with PBS containing 1% sodium dodecyl sulfate (SDS) (75 μL/well), and mixed with 225 µL/well of LB broth. After phage amplification in BLT5615, cDNA inserts of the enriched phages were amplified by PCR, purified (400–1500 bp) by agarose gel electrophoresis using Pippin Prep (Sage Science, Beverly, MA, USA), and sequenced by next-generation sequencing. Resulting short reads were assessed for quality using FastQ and aligned to the NCBI CCDS database with BWA. Read counts of identified sequences were compared between DR and healthy cells.

### 2.7. Statistical Analysis

Data are expressed as mean ± SEM. Intergroup differences were analyzed using a two-tailed Student’s *t*-test. Ligandomics data were normalized based on the total valid reads identified by next-generation sequencing and compared between diabetic and healthy cells using the Chi-square test [4]. Binding activity plots were utilized to visualize global changes in ligand binding activity [4]. Pearson correlation analysis was used to calculate correlation coefficients.

## 3. Results

### 3.1. Cell Purity of Immunopanned ECs and RGCs

Retinal ECs and RGCs are distinguished by their high expression of the transmembrane protein CD31 (PECAM-1) and CD90.2 (Thy-1.2), respectively, on the plasma membrane [14,15]. We purified these cells through immunopanning using plates pre-coated with anti-CD31 and anti-CD90.2 mAbs. To assess cell purity, we detached and analyzed ECs and RGCs by flow cytometry using anti-CD31 and anti-βIII-tubulin mAbs, respectively. The results showed that 99.3% of the immunopanned ECs were CD31-positive (Figure 1A), while commercially obtained HRMVECs as a positive control were 99.9% CD31-positive (Figure 1B). Additionally, flow cytometry revealed that 84.8% of immunopanned RGCs expressed βIII-tubulin (Figure 1C), with the HT22 neuronal cell line as a positive control showing 97.9% βIII-tubulin positivity (Figure 1D). These findings indicate that the immunopanned ECs and RGCs were of relatively high purity.

### 3.2. Binding of Scg3-Phage and VEGF-Phage to Immunopanned ECs

To determine whether isolated ECs exhibit binding activity patterns similar to those observed in vivo, we quantified their binding to Scg3 or VEGF using a ligand binding assay, as recently described [16]. Immunopanned ECs were incubated with clonal Scg3-Phage or VEGF-Phage in the presence or absence of their respective antagonists, Scg3-neutralizing hFab or aflibercept. After washing, cell-bound phages were eluted and quantified by a plaque assay. The results showed minimal binding of Scg3-Phage to healthy ECs, but a significant 7.1-fold increase in binding to diabetic ECs (Figure 2A). Anti-Scg3 hFab significantly blocked 62% of Scg3-Phage binding to diabetic ECs, whereas the blockade was not significant (15%) in healthy ECs. This binding activity pattern is consistent with Scg3 selective binding to DR but not healthy retinal vessels in vivo [10], suggesting that the ex vivo ligand binding assay accurately reflects in vivo ligand binding quantification.

VEGF-Phage binding to healthy ECs was also minimal, whereas its binding to diabetic ECs increased moderately by 3.3-fold, which is less than the increase observed for Scg3 binding to diabetic ECs (Figure 2B). Additionally, aflibercept blocked only 28% and 12% of VEGF-Phage binding to diabetic and healthy ECs, respectively (Figure 2B). The control phage showed minimal binding to healthy ECs (Figure 2A,B). These findings suggest that Scg3 selectively binds to DR ECs, whereas VEGF has significantly less binding to diabetic ECs.

### 3.3. Binding of Scg3-Phage and VEGF-Phage to Immunopanned RGCs

We further quantified the binding of Scg3-Phage and VEGF-Phage to immunopanned RGCs. There was no increase in Scg3 binding to diabetic RGCs, and no specific Scg3 binding was observed in the presence versus absence of anti-Scg3 hFab for either diabetic or healthy RGCs (Figure 3A). In contrast, VEGF binding to diabetic RGCs was 1.8-fold higher than to healthy RGCs. Aflibercept blocked 30% of VEGF binding to healthy RGCs and 44% to diabetic RGCs (Figure 3B), consistent with reports that VEGF functions as a neurotrophic factor [17].

### 3.4. Ex Vivo Ligandomics Profiling Using Immunopanned ECs

To evaluate the feasibility of ex vivo ligandomics, we performed ligandomics profiling on immunopanned ECs through three rounds of cell-binding selection. In brief, open reading frame phage display (OPD) cDNA libraries were purified and incubated with immunopanned diabetic and healthy retinal ECs, followed by the washing, elution, and quantification of total EC-bound phages. The eluted phage clones were then amplified, repurified, and used as inputs for three rounds of EC-binding selection to enrich cell-binding ligands. By Round 3, the binding of OPD cDNA libraries increased by 10.8-fold for healthy ECs and 16.9-fold for diabetic ECs (Figure 4A). At this stage, more phage clones bound to diabetic ECs compared to healthy ECs, consistent with our previous in vivo ligandomics profiling, which showed a higher number of phage library clones binding to diabetic retinal vessels than to healthy vessels [4]. 

Next-generation DNA sequencing identified 6,767,967 and 2,570,263 sequence reads, aligning to 2225 and 1624 genes for diabetic and healthy ECs, respectively. A binding activity plot comparing diabetic versus healthy ECs revealed a dense cluster of ligands with up to a 10-fold increase in binding to diabetic ECs, positioned above the diagonal line that represents minimal binding activity change (Figure 4B). Other ligands exhibited up to a 1000-fold increase in binding to healthy ECs with a relatively even distribution on the right side of the diagonal line. Pearson correlation analysis yielded a coefficient of 0.913, indicating moderate global changes in ligand binding activity. Using the same criteria as in our previous in vivo ligandomics profiling (*p* < 0.001 by Chi-square test; diabetes/control binding activity ratio ≥ 10 or ≤ 0.1; read counts of DR or control ≥ 30) [4], we identified 668 ligands with increased binding and 103 ligands with decreased binding to diabetic ECs.

Quantitative analysis showed that Scg3 was significantly enriched by 5.6-fold in diabetic ECs compared to healthy ECs (Figure 4C). These results confirm that Scg3 clones can be enriched by ex vivo ligandomics as a DR-selective EC ligand, consistent with our in vivo ligandomics profiling in diabetic versus healthy mice [4].

### 3.5. Ex Vivo Ligandomics Profiling Using Immunopanned RGCs

We also conducted ex vivo ligandomics profiling on immunopanned RGCs. The results revealed a 14.5-fold increase in binding to healthy RGCs and a 17.8-fold increase to diabetic RGCs (Figure 5A). By Round 3, more phage clones bound to diabetic RGCs compared to healthy RGCs.

Next-generation sequencing identified 9,941,846 and 5,393,498 sequence reads, corresponding to 1333 and 2426 genes for diabetic and healthy RGCs, respectively. A binding activity plot comparing diabetic versus healthy RGCs showed a dense cluster of ligands with about a 100- to 100-fold reduction in binding to diabetic cells, located on the right side of the diagonal line (Figure 5B). Other ligands exhibited up to a 100-fold increase in binding activity to diabetic RGCs with an evenly spread distribution. This pattern of global binding activity changes for diabetic RGCs contrasts with the opposite pattern observed for diabetic ECs (Figure 5B versus Figure 4B). Pearson correlation analysis resulted in a coefficient of 0.712, suggesting more pronounced global changes in ligand binding activity compared to immunopanned ECs (Figure 5B versus Figure 4B). Using the aforementioned criteria, we identified 53 and 1,644 ligands with significantly increased binding to diabetic and healthy RGCs, respectively.

Quantitative analysis revealed no preferential enrichment of Scg3 with diabetic RGCs, with only low background binding detected (Figure 5C). These findings confirm that Scg3 cannot be enriched by ex vivo ligandomics as a DR-selective RGC ligand.

## 4. Discussion

We have recently developed ligandomics technology for the global mapping of cell-binding ligands with the simultaneous quantification of binding activity [5,18]. When applied to anesthetized mice with or without DR and choroidal neovascularization, comparative ligandomics globally and quantitatively mapped disease-selective endothelial ligands [4,6]. Furthermore, we developed four functional assays—in vivo ligand binding, functional immunohistochemistry (FIHC), function-first, and therapy-first analyses—to independently validate identified ligands [6,8,10,11,19].

Scg3 has been independently validated as a disease-restricted angiogenic factor that selectively binds to and stimulates the angiogenesis of diseased but not healthy vessels [4,6,8,11,19]. In contrast, conventional angiogenic factors, such as VEGF, indiscriminately drive angiogenesis [4]. Consequently, Scg3-neutralizing antibodies selectively inhibit pathological but not physiological angiogenesis with minimal adverse side effects on healthy vessels, offering significant safety advantages over VEGF antagonists in preclinical studies [7,8,20].

Ligandomics uniquely integrates functional and quantitative capacities for target profiling [5]. While mass spectrometry-based functional proteomics efficiently maps protein interactomes, it falls short in quantifying protein–protein interaction binding activity [21]. Conversely, expression proteomics and transcriptomics, including single-cell RNA-seq (scRNA-seq), can globally and quantitatively map protein and gene expression under various conditions but cannot profile functional interactions [22,23]. To circumvent these limitations, computational tools like CellChat have been developed to infer cell–cell communication and ligand–receptor interactions (LRIs) from scRNA-seq data, offering broad applicability, high-resolution data, quantitative analysis, and multiple visualization methods [24,25]. However, these computational approaches have limited reliability to predict LRIs due to poor mRNA–protein correlation, non-coding RNA regulation, potential posttranscriptional and posttranslational modifications, protein trafficking and stability, unknown LRIs, and promiscuous LRIs [5,26,27,28]. In contrast, ligandomics excels among omics technologies by providing simultaneous quantitative and functional profiling capacities and experimentally quantifying ligand–cell binding activity for reliable ligand selection [4,6].

Despite the advantage, in vivo ligandomics necessitates systemic circulation for ligand binding and intracardial perfusion to remove unbound library clones, limiting its compatibility only to vascular ECs. These requirements significantly restrict the application of ligandomics to other cell types, including neural, immune, mesenchymal, epithelial, muscle, cancer, and stem cells.

This study investigates the feasibility of ex vivo ligandomics (Figure 6), which is broadly applicable to any type of isolated cells. We found that Scg3 selectively binds to immunopanned diabetic but not healthy ECs, consistent with our in vivo ligandomics profiling and ligand binding assay results [4,10]. Additionally, Scg3 shows minimal binding to both healthy and diabetic RGCs. This is supported by the minimal phenotype of Scg3^−/−^ mice, which are equivalent to a 100% blockade of Scg3 and exhibit no defects in neurogenesis and neuronal function [8,29]. Furthermore, VEGF binds to both healthy and diabetic RGCs (Figure 3B), consistent with its role as a neurotrophic factor [17]. While VEGF shows minimal binding to healthy ECs (Figure 2B), consistent with our in vivo ligand binding assay in mice [10], its binding moderately increases in diseased ECs (Figure 2B).

Unlike in vivo ligandomics, which is limited to vascular ECs in animal models, ex vivo can be applied to any cell types, tissues, diseases, and species, including human specimens. Human tissues surgically removed for clinical care are ideal for ex vivo ligandomics. Additionally, fresh tissues with a reasonable post-mortem interval can also be used for ligandomics profiling, providing cell viability is analyzed.

Our ex vivo ligand binding assay (Figure 2 and Figure 3) is a valuable approach for predicting on-target toxicity. For instance, VEGF binding to RGCs (Figure 3B) can help predict the on-target toxicity of anti-VEGF treatments on neurons. Indeed, VEGF inhibitors have been reported to cause neuronal side effects [30,31,32]. The minimal binding of Scg3 to RGCs in Figure 3A suggests negligible side effects of anti-Scg3 on neurons, consistent with the normal neural phenotype of Scg3^−/−^ mice and the safety profile of anti-Scg3 hFab [7,8,29]. This method can be extended to other disease-selective ligands identified by ligandomics. Before developing therapeutics, the binding of identified ligands to various cell types isolated from healthy and diseased tissues can be quantified using the ex vivo ligand binding assay to predict the spectrum of on-target toxicity. This will help optimize target selection and reduce safety-related attrition rates in drug research and development.

While this study utilizes immunopanning, fluorescence-activated cell sorting (FACS) can also be employed to isolate specific cell populations by labeling cell surface markers with fluorescence-conjugated antibodies. If cell-specific markers or antibodies are unavailable, genetically modified mice expressing green fluorescent protein or tdTomato in specific cell lineages can be used to sort cells via FACS [33].

Although this study confirms the feasibility of ex vivo ligandomics (Figure 6), cell isolation can alter the surface receptome and affect ligand binding properties. To minimize these changes, it is crucial to optimize isolation procedures by using collagenase or non-proteolytic hyaluronidase, instead of broad-spectrum proteases, such as trypsin or papain [34]. If necessary, isolated cells can be cultured briefly to allow the receptome to recover. However, the prolonged in vitro culture of isolated primary cells may result in the loss of their disease-related receptome phenotype. Therefore, these conditions should be carefully optimized for ex vivo ligandomics to sensitively and efficiently identify disease-selective ligands.

## 5. Conclusions

This study demonstrates the feasibility of ex vivo ligandomics using immunopanned cells (Figure 6). Since cells can be isolated from both healthy and diseased animal or human tissues, ex vivo ligandomics is broadly applicable to a wide range of cell types, tissues, diseases, and species.

## Figures and Tables

**Figure 1 biomolecules-15-00145-f001:**
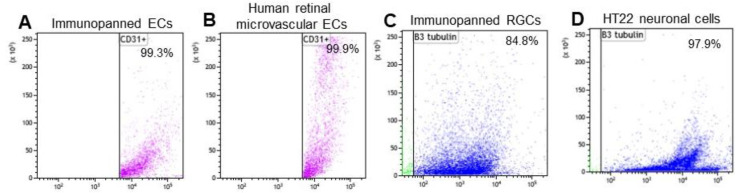
Flow cytometric analysis of the purity of immunopanned cells. Endothelial cells (ECs) and retinal ganglion cells (RGCs) were immunopanned using anti-CD31 and anti-CD90.2 mAb, respectively, detached by pipetting and analyzed by flow cytometry. (**A**) Immunopanned ECs were labeled with anti-CD31 mAb and quantified with 99.29% purity. (**B**) Human retinal microvascular endothelial cells (HRMVECs) were used as a positive control with 99.89% purity, as labeled with anti-CD31 mAb. (**C**) Immunopanned RGCs were labeled with anti-βIII-tubulin mAb and quantified with 84.80% purity. (**D**) HT22 neuronal cell line was used as a positive control with 97.87% purity, as detected by anti-βIII-tubulin mAb.

**Figure 2 biomolecules-15-00145-f002:**
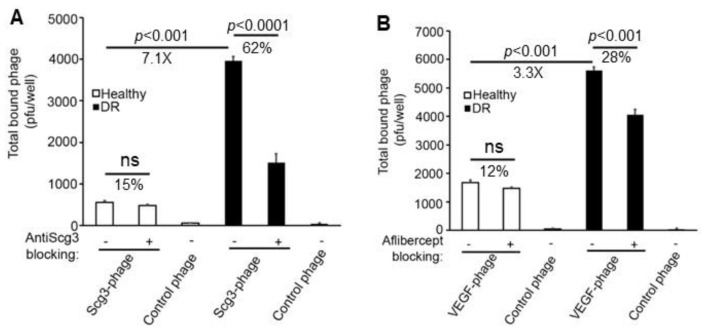
Ex vivo Scg3-Phage and VEGF-Phage binding to immunopanned CD31^+^ ECs. Cells were isolated from the retinas of healthy (white bar) and diabetic (black bar) mice. Control phage (*n* = 4 wells) was included as a negative control. (**A**) Binding of Scg3-Phage to ECs in the presence or absence of anti-Scg3 hFab (*n* = 5 wells/group). (**B**) Binding of VEGF-Phage to ECs in the presence or absence of aflibercept (*n* = 5 wells/group). ± SEM; ns, not significant; *t*-test.

**Figure 3 biomolecules-15-00145-f003:**
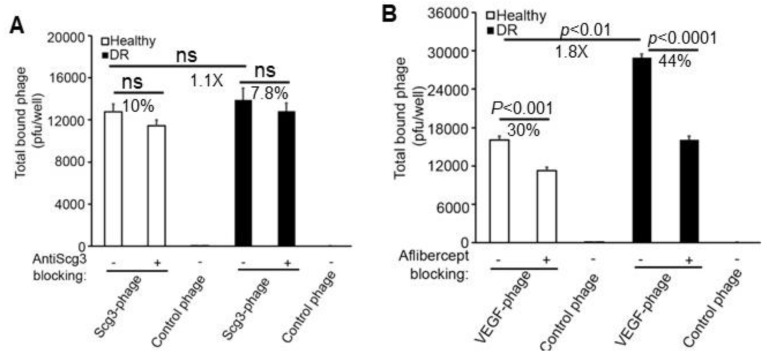
Ex vivo Scg3-Phage and VEGF-Phage binding to immunopanned CD90.2^+^ RGCs. Cells were isolated from the retinas of healthy (white bar) and diabetic (black bar) mice. Control phage (*n* = 4 wells) was included as a negative control. (**A**) Binding of Scg3-Phage to RGCs in the presence or absence of anti-Scg3 hFab (*n* = 5 wells/group). (**B**) Binding of VEGF-Phage in the presence or absence of aflibercept (*n* = 5 wells/group). ± SEM; ns, not significant; *t*-test.

**Figure 4 biomolecules-15-00145-f004:**
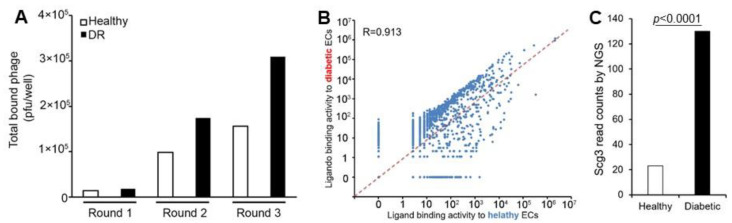
Ex vivo ligandomics profiling using immunopanned ECs. Cells were isolated from the retinas of healthy (white bar) and diabetic (black bar) mice by immunopanning. (**A**) Three rounds of binding selection were performed using immunopanned ECs for ex vivo ligandomics, followed by the quantification of cell-bound phages at each round. (**B**) The binding activity plot for diabetic versus healthy ECs visualizes the global changes in ligand binding activity. (**C**) Scg3 was significantly enriched by the binding selection to diabetic ECs, as quantified by next-generation sequencing. Chi square (χ^2^) test.

**Figure 5 biomolecules-15-00145-f005:**
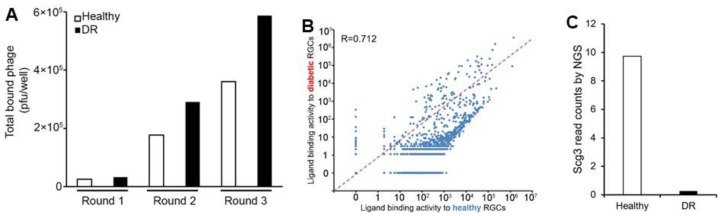
Ex vivo ligandomics profiling using immunopanned RGCs. Cells were isolated from the retinas of healthy (white bar) and diabetic (black bar) mice by immunopanning. (**A**) Three rounds of binding selection were performed using immunopanned RGCs for ex vivo ligandomics, followed by the quantification of cell-bound phages at each round. (**B**) The binding activity plot for diabetic versus healthy RGCs visualizes the global changes in ligand binding activity. (**C**) Scg3 was not enriched by the binding selection to diabetic RGCs, as quantified by next-generation sequencing. Chi square (χ^2^) test.

**Figure 6 biomolecules-15-00145-f006:**
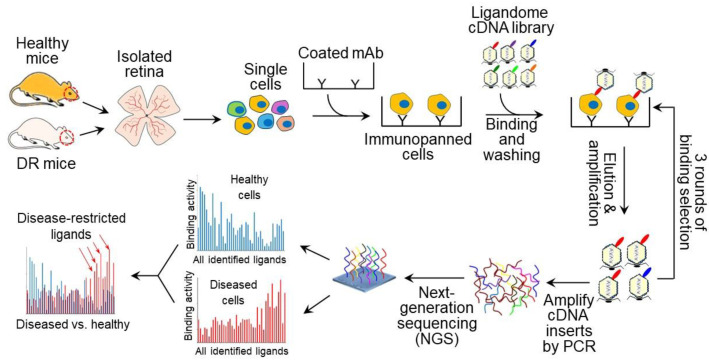
Schematic illustration of ex vivo ligandomics profiling. Retinas are isolated from mice with or without diabetic retinopathy (DR), separated into single cells, immunopanned to capture endothelial cells or retinal ganglion cells, and incubated with purified open reading frame phage display (OPD) cDNA libraries. After washing, cell-bound phages are eluted, quantified by a phage plaque assay, amplified, purified, and used as input for the next round of cell-binding selection. After three rounds of binding selection, cDNA inserts in the enriched phage clones are amplified by PCR and sequenced by next-generation sequencing, and the identified cell-binding ligands and their cDNA copy number are quantitatively compared to identify disease-selective cell-binding ligands.

## Data Availability

All data associated with this study are available upon request.

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
