# Peer review of "Feasibility of Ex Vivo Ligandomics"

_biomolecules, 2025, doi:10.3390/biom15010145_

Round 1

Reviewer 1 Report

Comments and Suggestions for Authors

This study by Waduge et al. reported the use of ligandomics technology in an ex vivo context, using diabetic retinopathy (DR) as a disease model. By isolating endothelial cells (ECs) and retinal ganglion cells (RGCs) from diabetic and healthy mice through immunopanning, the authors demonstrated that ex vivo ligand binding patterns closely mirror in vivo observations. The study highlights the potential of ex vivo ligandomics to map disease-selective ligands across diverse cell types and diseases, extending its utility beyond ECs. The authors considered ex vivo ligandomics as a solution to the limitations of in vivo approaches, particularly for non-vascular cell types. The integration of ligand binding assays, immunopanning, and next-generation sequencing ensured robust data acquisition and analysis. Furthermore, the validation of Scg3 as a disease-selective ligand for diabetic ECs underscores the clinical relevance of this method for developing targeted therapies.

This reviewer posits that the Discussion section include an analysis of the potential challenges in translating ex vivo ligandomics to human samples or other diseases. In the method, software tools for next-generation sequencing analysis, needs be explicitly stated to improve reproducibility. Lastly, few references lack uniform formatting and should be standardized for consistency.

Author Response

Reviewer 1:

Comment 1: Discussion section include an analysis of the potential challenges in translating ex vivo ligandomics to human samples or other diseases.

Response 1: Thanks for the suggestion. The potential challenges are discussed (Line 327 – 331)

Comment 2: In the method, software tools for next-generation sequencing analysis, needs be explicitly stated to improve reproducibility.

Response 2: The software tools are updated (Line 145-146).

Comment 3: Lastly, few references lack uniform formatting and should be standardized for consistency.

Response 3: Refer #3, 14, 25, 26 and 30 are updated.

Additionally, placing a common between the last name and the article title is not advisable, as this makes it difficult to determine where the title begins. Similarly, a comma at the end of the article title can cause confusion with the journal name. I have replaced these commons with periods to align with the reference style used in papers recently published in Biomolecules  (PMID: 39595563). 

Reviewer 2 Report

Comments and Suggestions for Authors

In this manuscript, the authors developed and validated ex vivo ligandomics that can be applied to various cell types and tissues,  diseases, and species. The work builds upon their previous research on in vivo ligandomics for profiling vascular ligands in mice.  Using the in vivo method, the authors previously discovered secretogranin III (Scg3) as a novel angiogenic factor that selectively binds to retinal vessels of diabetic but not healthy mice, leading to the development of anti-Scg3 therapy for ocular vasculopathies.

However, due to limitations of in vivo ligandomics which requires intracardial perfusion to remove unbound phage clones, restricting its use to vascular ECs, here the authors extended ligandomics to non-vascular cells.  They isolated ECs and retinal ganglion cells (RGCs) from diabetic and healthy mouse retinas by immunopanning, and confirmed that Scg3 and Vegf phages showed similar binding patterns as in vivo pattern, supporting the feasibility of ex vivo ligandomics.

The research is well-conducted, and the manuscript is well-written. Statistical analyses were correctly performed, and the data are presented effectively. The pros and cons of this method are thoroughly discussed.

The reviewer has only a few minor comments: 

1.     Line 213, please spell out OPD

2.  Consider giving brief details on how three rounds of cell-binding selection were performed in the result section

3.       Please consider adding statistics to Fig 4A

4.       Line 291, please spell out LRIs

5.       Please consider presenting Fig 6 as Fig 1 so the reader has an overall view of the experimental design at the beginning. 

Author Response

Reviewer 2:

Comment 1: Line 213, please spell out OPD

Response 1: Revised as suggested.  (Line 213)

Comment 2: Consider giving brief details on how three rounds of cell-binding selection were performed in the result section

Response 2: Thanks for the suggestion. We have added the brief details (Line 212 – 216).

Comment 3: Please consider adding statistics to Fig 4A.

Response 3: Fig. 4A and 5A represent intermediate steps in ligandomics profiling, analogous to total protein yields by affinity purification of protein interactomes in functional proteomics or total RNA yields in transcriptomics. These intermediate product quantities are seldom subjected to statistical analysis.

Comment 4: Line 291, please spell out LRIs

Response 4: Revised as suggested (Line 295).

Comment 5: Please consider presenting Fig 6 as Fig 1 so the reader has an overall view of the experimental design at the beginning.

Response 5: If a figure illustrates different protein domains or plasmid constructs, it would be more effective to place the structure figure at the beginning for easier reading. In contrast, Figure 6, which serves as a summary and integrates all the information, should be kept at the end. We hope this arrangement is acceptable to the reviewer.